# Deep learning for detecting retinal detachment and discerning macular status using ultra-widefield fundus images

Zhongwen Li[1,8], Chong Guo[1,8], Danyao Nie[3,8], Duoru Lin[1], Yi Zhu[1,4], Chuan Chen [1,4], Xiaohang Wu[1], Fabao Xu[1], Chenjin Jin[1], Xiayin Zhang[1], Hui Xiao[1], Kai Zhang [1,5], Lanqin Zhao[1], Pisong Yan[1], Weiyi Lai[1], Jianyin Li[1], Weibo Feng[1], Yonghao Li[1], Daniel Shu Wei Ting[6,7] & Haotian Lin [1,2]*

Retinal detachment can lead to severe visual loss if not treated timely. The early diagnosis of retinal detachment can improve the rate of successful reattachment and the visual results, especially before macular involvement. Manual retinal detachment screening is time-consuming and labour-intensive, which is difficult for large-scale clinical applications. In this study, we developed a cascaded deep learning system based on the ultra-widefield fundus images for automated retinal detachment detection and macula-on/off retinal detachment discerning. The performance of this system is reliable and comparable to an experienced ophthalmologist. In addition, this system can automatically provide guidance to patients regarding appropriate preoperative posturing to reduce retinal detachment progression and the urgency of retinal detachment repair. The implementation of this system on a global scale may drastically reduce the extent of vision impairment resulting from retinal detachment by providing timely identification and referral.

[1] State Key Laboratory of Ophthalmology, Zhongshan Ophthalmic Centre, Sun Yat-sen University, Guangzhou 510060, China. [2] Centre for Precision Medicine, Sun Yat-sen University, Guangzhou 510060, China. [3] Shenzhen Eye Hospital, Shenzhen Key Laboratory of Ophthalmology, Affiliated Shenzhen Eye Hospital of Jinan University, Shenzhen 518001, China. [4] Department of Molecular and Cellular Pharmacology, University of Miami Miller School of Medicine, Miami, Florida 33136, USA. [5] School of Computer Science and Technology, Xidian University, Xi'an 710071, China. [6] Singapore National Eye Centre, Singapore Eye Research Institute, 168751 Singapore, Singapore. [7] Duke-NUS Medical School, National University of Singapore, Singapore 119077, Singapore. [8] These authors contributed equally: Zhongwen Li, Chong Guo, Danyao Nie. *email: haot.lin@hotmail.com

Retinal detachment (RD) occurs when the sensory retina separates from the retinal pigment epithelium, which may lead to severe visual loss if not treated timely[1]. In Western countries, such as the United States, Scotland, and the Netherlands, the annual incidence of RD is 12.05 to 18.2 cases per 100,000 persons[2–5], with a peak incidence of 52.5 per 100,000 persons aged between 55 and 59 years[4]. In Eastern countries, such as China, Singapore, and Korea, the annual incidence of RD is 7.98 to 17.9 cases per 100,000 persons[6–10], with the highest incidence of 22.2 per 100,000 persons aged between 60 and 69 years[8].

Although RD often causes visual disability and visual loss, the early diagnosis of RD can improve the rate of successful reattachment and the visual results, especially before macular involvement[11]. However, identifying RD at an early stage remains challenging, as RD often starts asymptomatically and progresses from a small size at the peripheral retina[12]. Even in patients who perceive early RD symptoms, such as flashes, curtain, and increasing floaters, 17% of these patients attribute these changes to aging or problems with their glasses or contact lenses[13]. Therefore, patients rarely visit a doctor in a timely manner until the symptoms aggravate or visual acuity sharply drops[13]. As a result, the best timing of treatment can be missed and irreversible visual loss may occur. Consequently, it is essential to develop an appropriate screening approach to detect RD at an early stage.

The early detection of RD requires experienced ophthalmologists to examine the whole retina through a dilated fundus after mydriasis, which is time-consuming and labour-intensive. These challenges substantially hinder the implementation of RD screening, particularly in a large population or underdeveloped areas with few ophthalmologists. Recently, the development of artificial intelligence (AI) has enabled the efficient and automatic detection of retinopathies such as diabetic retinopathy (DR), age-related degeneration (AMD), and glaucoma. However, most of these studies trained the deep learning models using images acquired from traditional fundus camera imaging[14–19]. Such imaging is adequate for observation of the optic nerve and posterior pole, but provides little information regarding the peripheral retina due to the limited visible scope (30° to 60°).

To efficiently screen RD using AI, the prerequisite is to obtain fundus images covering the peripheral retina. The ultra-widefield fundus (UWF) imaging system meets this requirement by providing 200° panoramic images of the retina[20]. More importantly, the peripheral retina can be observed through UWF imaging with a single capture without requiring a dark setting, contact lens, or pupillary dilation[20]. The employment of UWF images in conjunction with deep machine learning algorithms may provide accurate identification of RD with high efficiency, thus facilitating the implementation of RD screening in the general population. Besides, the presence or absence of macular detachment is a major determinant of the urgency of RD repair and the visual prognosis after surgery, indicating that the macula-on RD patient needs a more urgent surgery and could have better visual acuity postoperatively than those with macula-off RD[1, 21, 22]. Therefore, in this study, we aimed to develop and evaluate a cascaded deep learning system for detecting RD and discerning the macular status based on UWF images.

## Results

**Baseline information.** As shown in Fig. 1, 11,087 UWF images from 7966 patients aged 10–86 years (mean age 47.5 years, 43.6% female) were labelled for RD. After deleting 636 poor-quality images due to the opacity of the refractive media or artifacts (e.g., arc defects, dust spots, and eyelash images), the first deep learning model designed to identify RD was developed using 10,451 images, 2009 of which were classified as RD, while the remaining 8442 images were classified as non-RD. All eligible images were randomly divided into 3 sets (no overlapping patients), with 70% (7323 images) as a training set, 15% (1556 images) as a validation

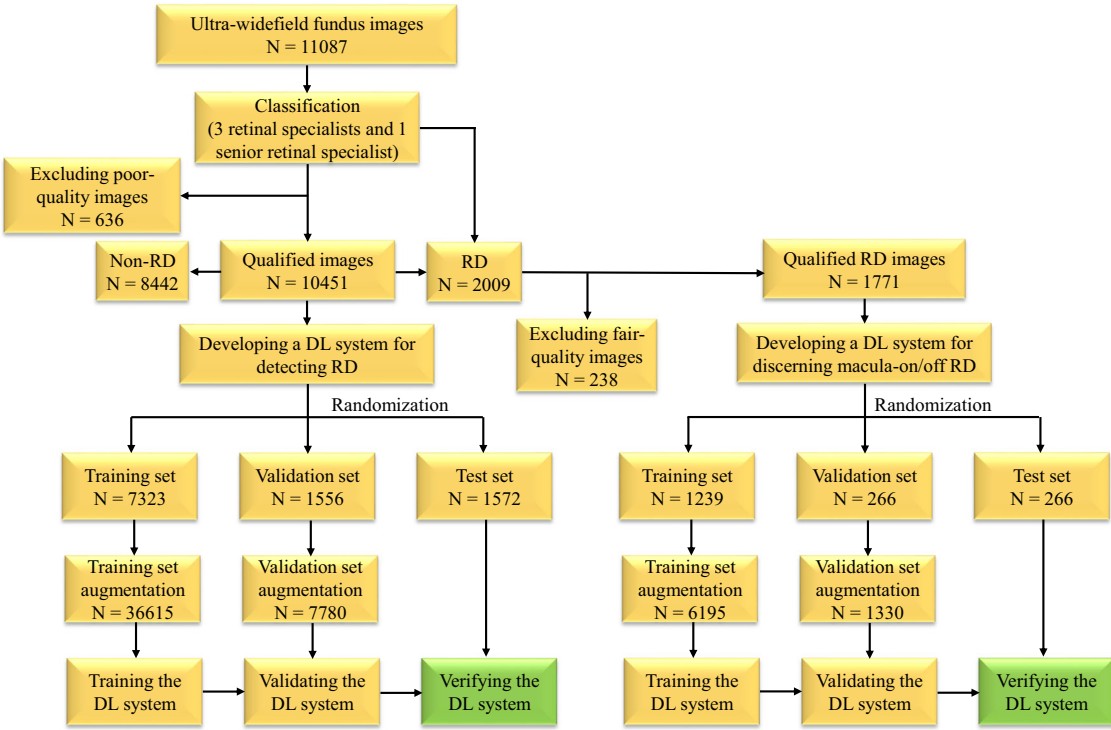

**Fig. 1** The workflow of developing deep learning (DL) systems for identifying retinal detachment (RD) and discerning macula-on/off RD based on ultra-widefield fundus images.

set, and 15% (1572 images) as a test set. After excluding 238 fair-quality images, 1771 RD images were used to build the second deep learning model designed to discern macula-on RD from macula-off RD, with 70% (1239 images) as a training set, 15% (266 images) as a validation set, and 15% (266 images) as a test set. The functions of the above two models are displayed in Fig. 2. In addition, the RD category in our study included images of the following five different types: rhegmatogenous RD, tractional RD, exudative RD, recurrent RD after scleral buckling, and recurrent RD after vitrectomy. The non-RD category included images of the normal retina and various retinopathies such as DR, AMD, and central serous chorioretinopathy. The numbers of labels in the training, validation, and test datasets are shown in Table 1.

**Performance of deep learning models**. The performance of the deep learning models and general ophthalmologists to detect RD and to discern the macular status is shown in Table 2. For RD detection, the general ophthalmologist with 5 years of experience had a 95.4% sensitivity and a 99.8% specificity, and the general ophthalmologist with 3 years of experience had a 94.4% sensitivity and a 99.1% specificity, whereas the first model had a 96.1% sensitivity and a 99.6% specificity with an area under the curve (AUC) of 0.989 (95% confidence interval (CI): 0.978–0.996) (Fig. 3a). For discerning macula-on RD from macula-off RD, the general ophthalmologist with 5 years of experience had a 91.3% sensitivity and a 92.4% specificity, and the general ophthalmologist with 3 years of experience had an 86.3% sensitivity and an 87.1% specificity, whereas the second model had a 93.8% sensitivity and a 90.9% specificity with an AUC of 0.975 (95% CI:

0.957–0.988) (Fig. 3b). Compared with the reference standard, the unweighted Cohen's κ coefficients were 0.965 (95% CI: 0.948–0.982) and 0.811 (95% CI: 0.736–0.887) for the first model and the second model, respectively. The performance of the models in the dataset from Zhongshan Ophthalmic Centre is displayed in Table 3.

**False-negative and false-positive findings**. Among the 12 RD images classified erroneously into the non-RD category by the first model, 2 showed localized RD surrounded by laser scars, 2 showed partial opacity of the refractive media, 5 presented very shallow RD, and 3 were too dark for an accurate assessment by the model (Fig. 4a). In contrast, five non-RD images were mistakenly assigned to the RD category, including one image of fundus albipunctatus, one image of retinal breaks, two images showing changes associated with subretinal proliferative membranes, and one image of a normal fundus including part of the conjunctiva of the lower eyelid (Fig. 4b). Similarly, using the second model, five images showed macula-on RD misclassified as macula-off RD. Among these images, two images showed an epiretinal membrane within the macula area, two showed macular distortion due to traction from the lesion in the peripheral retina, and one showed an atrophic macula (Fig. 5a). In contrast, of 17 macula-off RD images misclassified as macula-on RD images, 8 showed a strong light reflex within the region of the macula, 6 showed shallow macular detachment, and the remaining 3 showed exudative RD (Fig. 5b).

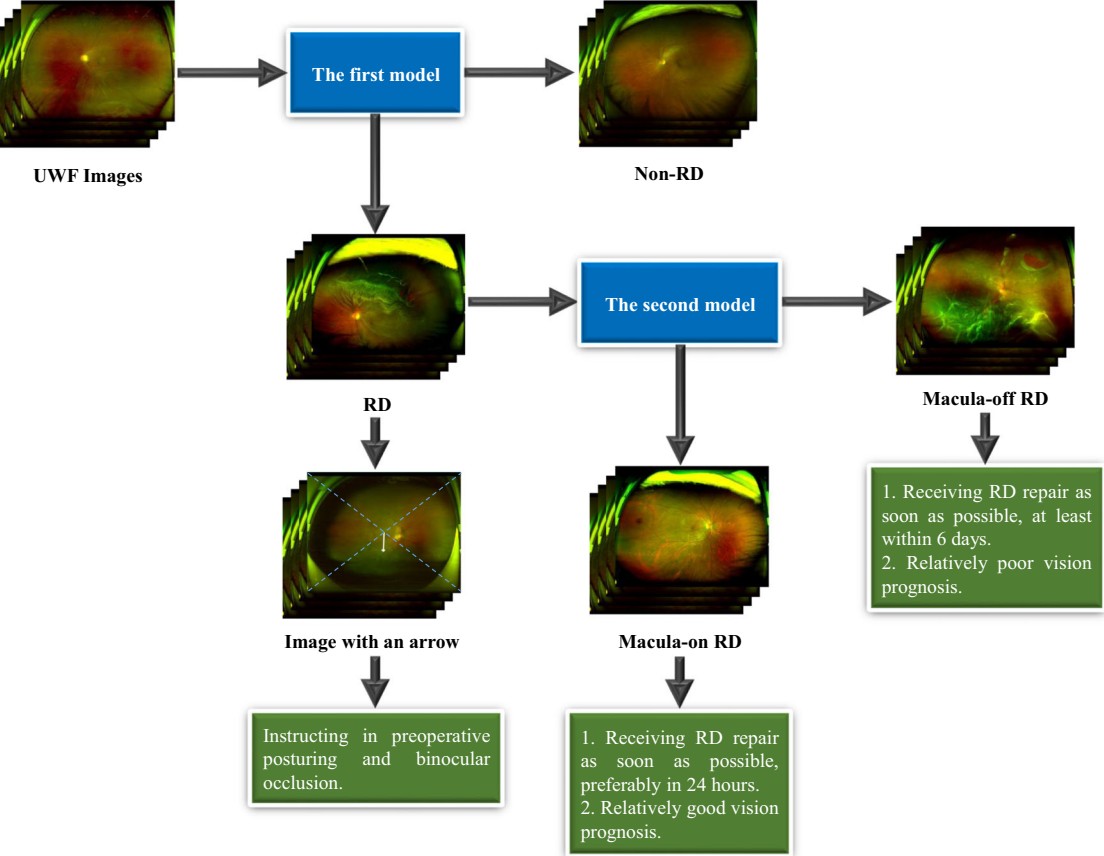

**Fig. 2 Framework of the cascaded deep learning system and its corresponding clinical application.** The first model is used to identify retinal detachment (RD) and the second model is used to identify macula-on RD. UWF ultra-widefield fundus.

**Medical guidance based on heatmaps.** In the test set, all 292 true-positive RD images displayed heatmap visualization in the region of RD. An arrow was successfully created and embedded in the UWF image automatically to instruct the RD patients in appropriate preoperative posturing, which could reduce RD progression after detection and before surgical treatment. For peripheral RD, the patients are instructed to turn their heads to the direction of the quadrant indicated by the arrow and maintain the position (Fig. 6a). For RD primarily located in the area of the macula or approximately total RD, a circle was displayed on the image, which indicated that the patients should maintain a supine position (Fig. 6b). In ten images, the arrows transformed into circles when the peripheral RD rather than the macula appeared at the centre of these images due to incorrect gazing during image capturing (Fig. 7).

## Discussion

In this study, we developed a cascaded deep learning system containing two models based on 11,087 UWF images. The first deep learning model was used to identify RD and showed robust performance (AUC 0.989, sensitivity 96.1%, and specificity 99.6%). The second deep learning model used to discern macula-on RD from macula-off RD also exhibited ideal performance (AUC 0.975, sensitivity 93.8%, and specificity 90.9%). In addition, the agreement between the deep learning models and the reference standard was almost perfect according to the unweighted

Cohen's κ coefficients (the first model 0.965, the second model 0.811). These results validate that our deep learning system provides an objective RD diagnosis with high accuracy and efficiency, while also determining whether the macula is involved. Besides, the sensitivities of both models were higher than those of the general ophthalmologists, although the general ophthalmologists with 5 years of experience had slightly higher specificities (Table 2). As high sensitivity is a prerequisite for a potential screening tool and can reduce the workload and medical costs by avoiding the need for further examination of evidently normal eyes[15, 23, 24], this system can be used to screen RD as a part of ophthalmic health evaluations in physical examination centres or community hospitals lacking ophthalmologists, or be deployed in hospitals with a large number of patients to assist ophthalmologists. In addition, this system can be applied to detect peripheral RD in patients who cannot tolerate a dilated fundus examination, such as those with a shallow peripheral anterior chamber. More importantly, based on the images obtained from a different UWF imager at another institution, the models still performed well for RD detection and macula-on RD recognition, which proves the generalizability of our deep learning system (Table 3). Previously, Ohsugi et al.[25] reported a deep learning-based diagnostic system for rhegmatogenous RD using a small number of UWF images ($n = 831$) without external validation. Compared with their study, the system developed in this study can identify different kinds of RD based on 10451 UWF images and has verified its efficacy in an independent external dataset, which is more representative of the real world.

In clinics, interpreting the output of a deep learning system is important for clinicians' and patients' acceptance of the system[26]. Thus, we performed heatmap visualization in all 292 true-positive RD images to identify the regions contributing the most to the deep learning system's classification. Inspiringly, the result showed that all these images highlighted the region of RD, which further corroborated the validity of our deep learning system. Similarly, Kermany et al.[27] performed an occlusion test on 491 optical coherence tomography images to identify the most important area that the neural network used to predict AMD and this test identified the region of interest in 94.7% of the images and correctly located drusen in 100% of all the images.

Preoperative posturing on the side where the RD mainly located is an effective strategy to reduce the progression of macula-on RD[28, 29]. However, in our view, although macula detachment occurs in partial RD, preoperative posturing is still indispensable as the extent of detachment is positively related to the failure rate of RD repair[30]. In addition, bed rest is proposed to RD patients, as this strategy can reduce the quantity of subretinal fluid, which can facilitate the performance of surgery and lower the complication rate[31]. Therefore, in this study, an arrow used to instruct the RD patients in preoperative positioning between

**Table 1 Number of ultra-widefield fundus images in the training, validation, and test datasets.**

|  | Training set no. (%) | Validation set no. (%) | Test set no. (%) |
|---|---|---|---|
| Rhegmatogenous RD | 1169 (16.0) | 244 (15.9) | 256 (16.3) |
| Tractional RD | 50 (0.7) | 15 (1.0) | 11 (0.7) |
| Exudative RD | 54 (0.7) | 12 (0.8) | 7 (0.4) |
| Recurrent RD after scleral buckling | 59 (0.8) | 14 (0.9) | 16 (1.0) |
| Recurrent RD after vitrectomy | 72 (1.0) | 16 (1.0) | 14 (0.9) |
| RDs | 1404 (19.2) | 301 (19.3) | 304 (19.3) |
| Non-RD | 5919 (80.8) | 1255 (80.7) | 1268 (80.7) |
| Total images of RD/non-RD | 7323 (100) | 1556 (100) | 1572 (100) |
| Macula-on RD | 360 (29.1) | 77 (28.9) | 80 (30.1) |
| Macula-off RD | 879 (70.9) | 189 (71.1) | 186 (69.9) |
| Total images of macula-on/off RD | 1239 (100) | 266 (100) | 266 (100) |

*RD* retinal detachment

**Table 2 Performance of the deep learning models vs. general ophthalmologists in the test sets.**

|  | Sensitivity (95% CI) | Specificity (95% CI) | Accuracy (95% CI) |
|---|---|---|---|
| Retinal detachment |  |  |  |
| Ophthalmologist A | 95.4% (93.0–97.8) | 99.8% (99.6–100) | 99.0% (98.5–99.5) |
| Ophthalmologist B | 94.4% (91.7–97.1) | 99.1% (98.6–99.6) | 98.2% (97.5–98.9) |
| The first model | 96.1% (93.9–98.3) | 99.6% (99.3–99.9) | 98.9% (98.4–99.4) |
| Macula-on detachment |  |  |  |
| Ophthalmologist A | 91.3% (84.8–97.8) | 92.4% (88.4–96.4) | 92.1% (88.7–95.5) |
| Ophthalmologist B | 86.3% (78.2–94.4) | 87.1% (81.9–92.3) | 86.8% (82.4–91.2) |
| The second model | 93.8% (88.3–99.3) | 90.9% (86.6–95.2) | 91.7% (88.2–95.2) |

A general ophthalmologist with 5 years of working experience at a physical examination centre, B general ophthalmologist with 3 years of working experience at a physical examination centre, CI confidence interval

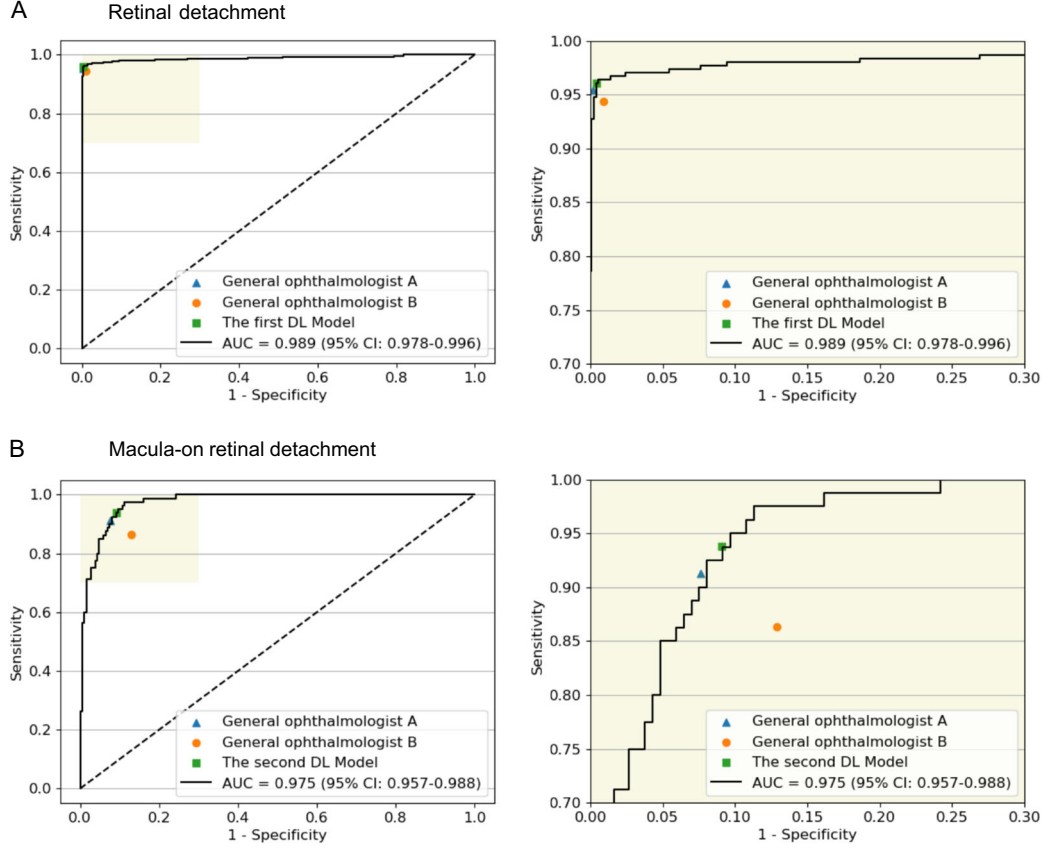

**Fig. 3 Receiver operating characteristic (ROC) curves of the deep learning models derived from the test datasets, compared with general ophthalmologists' performance using reference standard. a** Detection performance of retinal detachment. **b** Detection performance of macula-on retinal detachment. AUC area under the ROC curve; General ophthalmologist A, 5 years of working experience at a physical examination centre; General ophthalmologist **b**, 3 years of working experience at a physical examination centre.

**Table 3 Performance of the deep learning models in the dataset from the Zhongshan Ophthalmic Centre.**

|  | AUC | Sensitivity (95% CI) | Specificity (95% CI) | Accuracy (95% CI) |
|---|---|---|---|---|
| The first model | 1.000 (95% CI: 0.999–1.000) | 99.5% (98.5–100) | 99.5% (98.5–100) | 99.5% (98.8–100) |
| The second model | 0.934 (95% CI: 0.846–0.995) | 93.5% (84.5–100) | 95.3% (92.0–98.6) | 95.0% (91.9–98.1) |

*AUC* area under the curve, *RD* retinal detachment. The first model is used to identify RD. The second model is used to identify macula-on RD

diagnosis and treatment was automatically created and embedded in the images through heatmaps (Fig. 6). On the premise of bed-rest immobilization, patients with an upward arrow in the images (RD mainly in the superior quadrant) should be positioned supine with their head bending back; patients with a leftward arrow (RD mainly in the left quadrant) should be positioned on the left side; patients with a rightward arrow (RD mainly in the right quadrant) should be positioned on the right side; patients with a downward arrow (RD mainly in the inferior quadrant) should be instructed to maintain a semi-supine position or sit upright; and patients with a circle (RD mainly in the macula area or total RD) should be instructed to maintain the supine position. Meanwhile, for the RD patients identified by our deep learning system, binocular occlusion is advised, because this strategy can decrease the frequency and amplitude of eye movements[32]. In combination with posturing, binocular occlusion can reduce the size of RD or at least prevent its further spreading[32, 33].

On the basis of the high accuracy in discerning macular-on/off RD, our deep learning system could be applied to notify patients

of the optimal timing of surgery and the possible visual prognosis after surgery. According to the guidelines from the American Academy of Ophthalmology, RD repair should be performed on patients with attached macula prior to patients with detached macula as soon as possible[1]. Specifically, for macula-on RD, surgical treatment should be performed before the macula is involved, preferably within 24 h; for macula-off RD, surgical treatment should be conducted within 6 days[1, 34, 35]. In addition, eyes with macula-on RD have an 82–87% chance of retaining a visual acuity of 20/50 or better postoperatively, whereas eyes with macula-off RD only have a 20–37% chance of achieving such visual acuity[36–38]. Therefore, our system not only facilitates the detection of RD, but also may help to greatly improve and predict the prognosis of RD in the clinic by discerning the macular status.

For better recognition of RD regardless of pathogenesis, our study included three different types of RD (rhegmatogenous RD, exudative RD, and tractional RD) in the datasets to develop and evaluate the deep learning system (Table 1). Moreover, to increase the application range of our system, recurrent RD after scleral

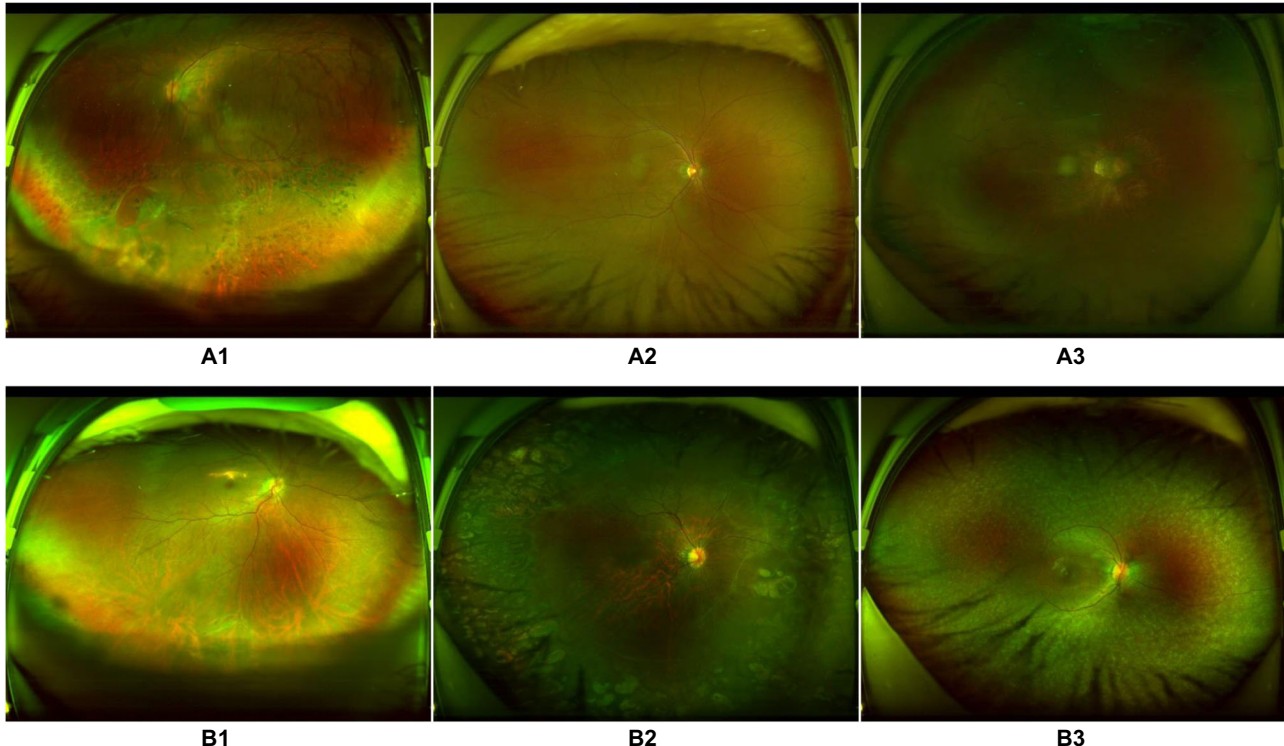

**Fig. 4 Ultra-widefield fundus images showing typical misclassified cases in retinal detachment (RD) detection. a** False-negative images: A1, RD surrounded by laser scars on the bottom left; A2, shallow RD at the bottom; A3, low-brightness image with RD on the top right. **b** False-positive images: B1, retinal breaks at the bottom; B2, subretinal membrane on the right side; B3, fundus albipunctatus.

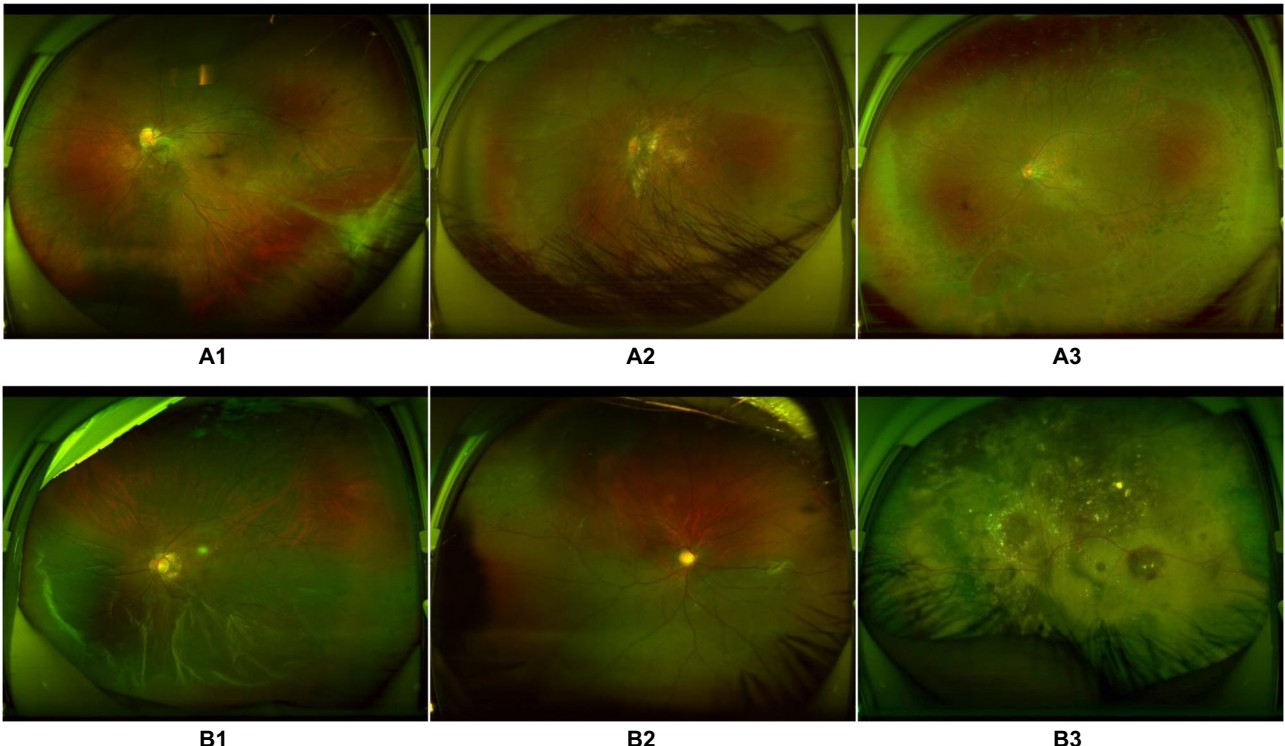

**Fig. 5 Ultra-widefield fundus images showing typical misclassified cases in macula-on retinal detachment (RD) detection. a** False-negative images: A1, macula-on RD with the distorted macula; A2, macula-on RD with an atrophic macula; A3, macula-on RD with an epiretinal membrane within the macular area. **b** False-positive images: B1, macula-off RD with a strong light reflex within the region of the macula; B2, shallow macular detachment; B3, exudative retinal detachment.

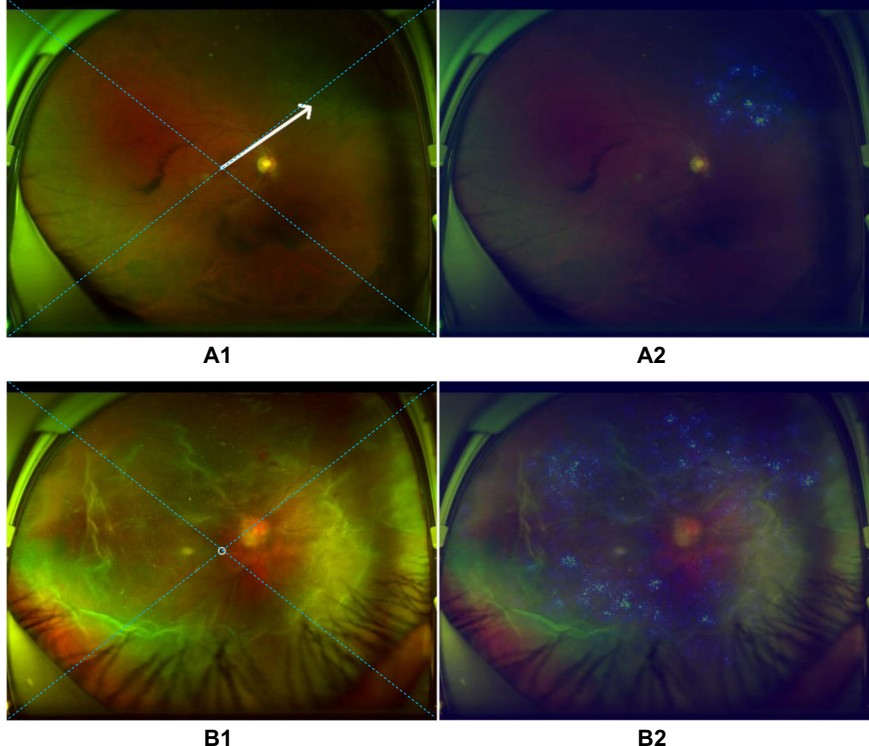

**Fig. 6 Examples of ultra-widefield fundus images with an arrow/circle generated according to heatmaps. a** The arrow towards the area of retinal detachment in image A1 is established automatically based on the highlighted region in heatmap A2. **b** The circle located at the centre of retinal detachment in image B1 is created automatically on the basis of the highlighted region in heatmap A2. The arrow/circle is used to instruct patients in preoperative posturing to reduce the progression of retinal detachment between detection and treatment. The dotted diagonal line in the image is used to divide the retina into four quadrants (superior, inferior, left, and right).

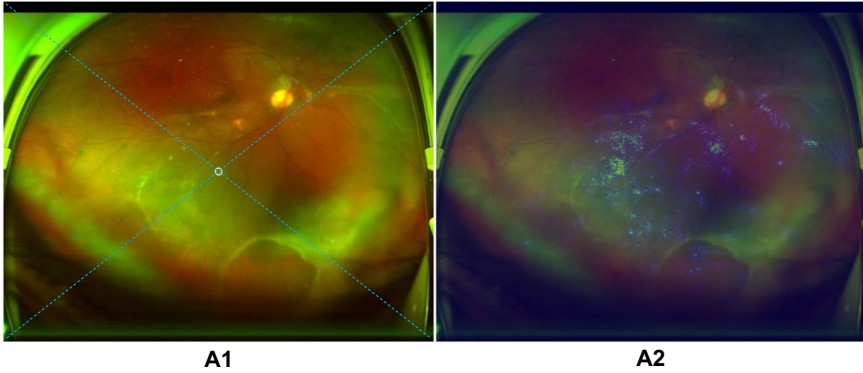

**Fig. 7 Example of an ultra-widefield fundus image with a mistaken circle generated according to a heatmap.** A circle which should have been a downward arrow in image A1 of inferior retinal detachment is established mistakenly based on the highlighted region in heatmap A2. The arrow/circle is used to instruct patients in preoperative posturing to reduce the progression of retinal detachment between detection and treatment. The dotted diagonal line in the image is used to divide the retina into four quadrants (superior, inferior, left, and right).

buckling or vitrectomy were also included in our datasets (Table 1). As images of recurrent RD from the test set were detected by the system, we speculated that our system could also be applied to RD recurrence detection.

Compared with ground truth, our deep learning models made a few mistakes, although they still had high accuracy. For the first model, when assessing the reasons for false-negative classification, approximately half of misclassified images were attributable to RD that was too shallow to be detected (41.7%). The remaining false-negative cases were a result of interference factors caused by opaque refractive media, dark image illumination, or laser scars. While evaluating the reasons for the false-positive classification, the features of all misclassified cases were similar to those of RD in varying degrees. For the second model, macula-on RD misclassified as macula-off RD was due to coexisting maculopathy. Conversely, among the reasons for macula-off RD misclassified as macula-on RD, almost half of misclassed images were caused by the strong light reflex in the area of the macula (47.1%) and the rest of the misclassified images were a result of shallow macular detachment. In the ten images with erroneous circles, the

peripheral RD was located at the centre of the picture instead of the macula due to an incorrect gazing direction during image capture, which led to the heatmap presenting in the centre. Thus, the circles, which should have been arrows, were mistakenly presented in the images (Fig. 7). To further improve the performance of our models, we intend to increase the number of these error-prone images in the training set in the future, which may potentially minimize both false-positive and false-negative results. To increase the accuracy of the icons (arrows/circles) in the RD images, the macula should be centred when taking a UWF photograph.

Several limitations exist in this study. First, although UWF imaging can capture the largest retinal view compared with other existing technologies, this method still could not cover the whole retina. Hence, our deep learning system may miss a few RD diagnoses that are not captured by UWF imaging. Moreover, a missed diagnosis would occur if RD appears in an obscured area of the image. Second, these images were collected from a tertiary hospital. Further research is needed to investigate the performance of the deep learning systems in other hospitals at different levels. In addition, financial support from the government or lowering the price of OPTOS imaging will aid in the deployment of deep learning-based OPTOS imaging generally, even in primary eye care settings.

In conclusion, the present study verifies that our robust cascaded deep learning system can be applied to identify RD and discern macula-on/off RD. Furthermore, our system can provide guidance to patients regarding appropriate preoperative posturing to reduce RD progression and the urgency of RD repair. Prospective clinical studies to evaluate the cost-effectiveness and the performance of this system in the real-world settings are ongoing.

## Methods

To develop the cascaded deep learning system, a total of 11,087 UWF images were retrospectively obtained from patients presenting for retinopathy examinations or undergoing a routine ophthalmic health evaluation between November 2016 and January 2019 at Shenzhen Eye Hospital using an OPTOS nonmydriatic camera (OPTOS Daytona, Dunefermline, UK) and 200° fields of view. Patients underwent this examination without mydriasis. All images were deidentified prior to transfer to research investigators. This study was approved by the Institutional Review Board of Zhongshan Ophthalmic Centre (Guangzhou, Guangdong, China) and adhered to the tenets of the Declaration of Helsinki.

**Image classification and the reference standard**. First, all images were classified into two types: RD and non-RD. Then, we further classified RD images into macula-on RD and macula-off RD according to whether subretinal fluid extended to involve the fovea[39]. Image quality was included in the classification as follows:

1. Excellent quality referred to images without any problems.
2. Good quality referred to images with deficiencies in focus, illumination, or other artifacts, but the region of RD and the macula could still be identified.
3. Fair quality referred to images with an obscured view of the macula, but part of the RD region could still be identified.
4. Poor quality referred to images that were insufficient for any interpretation (an obscured area over one-third of the image).

Poor-quality images were excluded from the study and fair-quality images were excluded from the training of the deep learning model designed to discern macula-on RD from macula-off RD. Figure 1 illustrates the workflow of image classification.

Training a deep learning system requires a robust reference standard[40, 41]. To ensure the accuracy of the image classification, all anonymous images were classified separately by three board-certified retinal specialists with at least 5 years of experience. The reference standard was determined based on the agreement achieved by all three retinal specialists. Any level of disagreement was arbitrated by another senior retinal specialist with over 20 years of experience. The deep learning system performance for identifying RD and discerning macula-on RD was compared against this reference standard.

**Image preprocessing and augmentation**. We performed image standardization before deep feature learning. First, the images were downsized to 512 by 512 and the pixel values were normalized within the range of 0 to 1. Data augmentation was applied to increase the diversity of the dataset and thus reduce the chance of overfitting

in the deep learning process. Horizontal and vertical flipping, rotation up to 90°, and brightness shift within the range of 0.8–1.6 were randomly applied to the images in the training and validation datasets to increase their size to five times the original size.

**Deep learning system development**. In the present study, the cascaded deep learning system included two models, each of which was trained using a state-of-the-art convolutional neural network (CNN) architecture InceptionResNetV2. InceptionResNetV2 mimics the architectural characteristics of two previous state-of-the-art CNNs: the Residual Network and the Inception Network. Weights pretrained for ImageNet classification were used to initialize the CNN architectures[42]. The first model identified RD in input images. The second model further discerned macula-on RD from RD images detected by the first model. Figure 2 shows the framework of this cascaded deep learning system.

Each deep learning model was trained up to 180 epochs. During the training process, the validation loss was evaluated using the validation set after each epoch and used as a reference for model selection. Early stopping was applied, and if the validation loss did not improve over 60 consecutive epochs, the training process was stopped. The model state where the validation loss was the lowest was saved as the final state of the model.

To train the first model, the whole dataset obtained from Shenzhen Eye Hospital was randomly divided into three independent sets: 70% in a training set, 15% in a validation set, and the remaining 15% in a test set, with no patients overlapping among these sets. To train the second model, the images of RD were randomly divided into three independent sets in the same manner. The training and validation sets were used to train and determine the models, respectively. The test sets were used to evaluate the performance of the selected models.

To verify the deep learning system, we randomly selected 400 qualified images (200 RD images and 200 non-RD images) from 383 patients, which were obtained from another type of OPTOS nonmydriatic camera (OPTOS 200Tx, Dunefermline, UK) in the Zhongshan Ophthalmic Centre between January and June 2019, to compare our system's classifications with those of the reference standard. Among 200 RD images, 31 are macula-on RD and 169 are macula-off RD.

**Characteristics of misclassification**. The UWF images, misclassified by the two deep learning models, were arbitrated and analysed by a senior retinal specialist, with the features documented.

**Visualization heatmap and arrow embedding**. To highlight the regions on which the deep learning system focused the most when detecting RD, heatmaps were generated using the Saliency Map visualization technique for all true-positive images. The Saliency Map technique calculates the gradient of the output of the CNN with respect to each pixel in the image, to identify the pixels with the greatest impact on the final prediction. To provide guidance to patients regarding head positioning, which may reduce RD progression before visiting their ophthalmologists, the centre of RD was located according to the hot regions in the heatmap and then an arrow pointing from the centre of the image to the centre of RD was automatically generated. Dotted diagonals were automatically created on the images to divide the retina into four quadrants (superior, inferior, left, and right). The specific direction of the preoperative posturing was determined according to the quadrant to which the arrow pointed. A circle was displayed on an image when the centre of the hot regions was within a 400-pixel diameter at the image centre, which was approximately equivalent to the area of the macula (overall image pixel size: 2600 × 2048).

**General ophthalmologist comparisons**. To evaluate our deep learning system in the context of screening RD, we recruited two general ophthalmologists who had 3 and 5 years of experience, respectively, in UWF image analysis at a physical examination centre, and then compared the performance of the deep learning models with that of the general ophthalmologists based on the test sets.

**Statistics and reproducibility**. We used a receiver operating characteristic curve and the AUC with 95% CIs to evaluate the performance of the deep learning models. The accuracy, sensitivity, and specificity of the models and general ophthalmologists for detecting RD and discerning the macular status were computed according to the reference standard. In the test sets, unweighted Cohen's κ coefficients were employed to compare the results of the models with the reference standard as determined by the aforementioned retinal specialists. All statistical analyses were performed using Python 3.7.3. The performance of the system can be repeated when training on the same UWF images with the same deep learning algorithm.

**Reporting summary**. Further information on research design is available in the Nature Research Reporting Summary linked to this article.

## Data availability

The datasets generated and/or analysed during the current study are available from the corresponding author on a reasonable request. Correspondence and requests for data materials should be addressed to HTL (haot.lin@hotmail.com). All datasets are stored at Baidu Netdisk.

## Code availability

The code used in this study can be accessed at GitHub (https://github.com/gocai/uwf_retinal_detachment).

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

## Acknowledgements

This study received funding from the National Key R&D Program of China (grant number 2018YFC0116500), the National Natural Science Foundation of China (grant number 81770967), the National Natural Science Fund for Distinguished Young Scholars (grant number 81822010), the Science and Technology Planning Projects of Guangdong Province (grant number 2018B010109008), and the Key Research Plan for the National Natural Science Foundation of China in Cultivation Project (grant number 91846109). The sponsor or funding organization had no role in the design or conduct of this research.

## Author contributions

Conception and design: Z.L., C.G., D.N., and H.L. Administrative support: H.L. Provision of study materials or patients: D.N. and Y.L. Collection and assembly of data: Z.L., D.L., X.W., F.X., X.Z., H.X., P.Y., W.L., J.L., and W.F. Data analysis and interpretation: Z.L., C.G., D.N., X.W., F.X., H.L., D.L., Y.Z., C.C., C.J., K.Z., D.T., P.Y., H.X., L.Z., W.L., and Y.L. Manuscript writing: all authors. Final approval of manuscript: all authors.

## Competing interests

The authors declare no competing interest.
