## [Peer Review File · Communications Biology]

Reviewers' comments:

Reviewer #1 (Remarks to the Author):

Authors developed deep learning based diagnostic system for retinal detachment.

"Accuracy of deep learning, a machine-learning technology, using ultra-wide-field fundus ophthalmoscopy for detecting rhegmatogenous retinal detachment. Ohsugi H, Tabuchi H, Enno H, Ishitobi N. Sci Rep. 2017 Aug 25;7(1):9425. doi: 10.1038/s41598-017-09891-x. PMID: 28842613" should be referred, because it is a previous study developing deep learning based diagnostic system for retinal detachment using ultra-wide-field funds ophthalmoscopy.

Despite the past study report, this study has new technology about discerning macular status. The macular status is important for RD patients.

Reviewer #2 (Remarks to the Author):

The manuscript entitled "A Cascaded Deep Learning System for Detecting Retinal Detachment and Discerning Macular Status Using Ultra-widefield Fundus Images" by Li et al describes a cascaded deep learning (DL) system based on the ultra-widefield fundus (UWF) images for automated RD detection and macula-on/off RD discerning. The results are interesting and conclusions are generally supported. However, there are several issues that should be addressed.

1.The authors try to develop an artificial intelligence RD detection system based on fundus images to help early diagnosis of RD. They choose ultra-widefield fundus (UWF) images because of its wide visible scope without pupillary dilation. But UWF imaging is not popular in clinic, and rare for routine medical examination. These will limit the potential applications of this system on helping early diagnosis of RD. Did the authors consider of using different kinds of fundus photos from several fundus cameras to build their screening system?

2.The RD detection model was developed using 10451 images, while the second model was developed from 1771 RD images. Such numbers of images maybe too small.

3.For the image data pre-processing, the images were downsized to 512 by 512. Could the authors explain how these 512 by 512 were chosen?

4.In line 311, "To verify the DL system, we randomly selected 400 qualified images (200 RD images and 200 non-RD images) from 383 patients". Is this part of 11087 UWF images, or some more data from other sources? Could the authors explain?

5.The authors categorize central serous chorioretinopathy as non-RD. But usually CSC manifested as shallow detachment of the macula. Did the model have difficult to identify CSC from RD or macula-off RD?

6.From figure 4, 5, 6, some images have shadow of eyelashes or eye lid. Did these shadows affect the model? How to minimize the interference ?

7.The authors concluded that "this system can optimal timing for RD repair, and predict the potential vision prognosis". This conclusion can't be achieved from one-time fundus image without follow-up. The authors should amend their conclusion.

A Cascaded Deep Learning System for Detecting Retinal Detachment and Discerning Macular Status Using Ultra-widefield Fundus Images

Response to the comments of reviewer #1

Thank you very much for your help! We sincerely appreciate your scrupulous and constructive suggestions, which are valuable for improving the quality of our article.

Comment 1: "Accuracy of deep learning, a machine-learning technology, using ultra-wide-field fundus ophthalmoscopy for detecting rhegmatogenous retinal detachment. Ohsugi H, Tabuchi H, Enno H, Ishitobi N. Sci Rep. 2017 Aug 25;7(1):9425. doi: 10.1038/s41598-017-09891-x. PMID: 28842613" should be referred, because it is a previous study developing deep learning based diagnostic system for retinal detachment using ultra-wide-field funds ophthalmoscopy.

Reply 1: Thank you for your advice. We referred and compared that study with our study as follows: Previously, Ohsugi et al reported a DL-based diagnostic system for rhegmatogenous RD using a small number of UWF images (n = 831) without external validation. Compared to their study, this study developed a DL system based on 10451 UWF images for identification of different kinds of RD, and verified its efficacy in an independent external dataset, which is more representative of the real world.

Changes in the text: See line 166 to 170.

Response to the comments of reviewer #2

Thank you very much for your help! We sincerely appreciate your scrupulous and constructive suggestions, which are valuable for improving the quality of our article.

Comment 1: The authors try to develop an artificial intelligence RD detection system based on fundus images to help early diagnosis of RD. They choose ultra-widefield fundus (UWF) images because of its wide visible scope without pupillary dilation. But UWF imaging is not popular in clinic, and rare for routine medical examination. These will limit the potential applications of this system on helping early diagnosis of RD. Did the authors consider of using different kinds of fundus photos from several fundus cameras to build their screening system?

Reply 1: Thank you for your questions. As the peripheral retina has drawing more attention by

ophthalmologists, in the past few years, since 2016, UWF imaging has become more popular. The reason is that many retinopathies begin at peripheral retina, such as retinal detachment and Eales disease. If we could detect them and prevent such diseases from further deterioration at early stage, it will be very helpful to reduce the visual impairment caused by these diseases. Moreover, routine exams were uncomfortable and time-consuming, especially for a child, which made it impossible for the doctor to conduct a complete exam and view the entire retina. In China, over 500 institutions of different sizes have UWF cameras now. In other countries, such as the UK, the number of hospitals which own the UWF cameras is also increasing. Therefore, on current trends, we believe our system will be more and more useful in the future.

We didn't use traditional fundus images to build our system as such images provide little information regarding the peripheral retina due to the limited visible scope (30° to 60°).

Comment 2: The RD detection model was developed using 10451 images, while the second model was developed from 1771 RD images. Such numbers of images maybe too small.

Reply 2: Thank you for your comments. The annual incidence of RD is around 20 cases per 100,000 persons. According to the criteria of WHO, it is not a common disease. That is why the RD images is not as many as non-RD images, but this sample size is the largest one when compared to previous RD studies, such as “Ohsugi H, Tabuchi H, Enno H, Ishitobi N. Sci Rep. 2017 Aug 25;7(1):9425. doi: 10.1038/s41598-017-09891-x. PMID: 28842613”. In addition, the second DL model performed well in the external dataset (accuracy = 95%).

Comment 3: For the image data pre-processing, the images were downsized to 512 by 512. Could the authors explain how these 512 by 512 were chosen?

Reply 3: Thank you for your question. For most deep learning tasks, images are downsized to fit the limited GPU memory. There are no strict rules to what this size should be, as long as it's large enough for the CNN. Some choose 299*299, some choose 256*256. We want to retain as much information as possible, while not overflowing the GPU memory, so we choose a slightly larger size 512*512.

Comment 4: In line 311, “To verify the DL system, we randomly selected 400 qualified images (200 RD images and 200 non-RD images) from 383 patients”. Is this part of 11087 UWF images, or some more data from other sources? Could the authors explain?

Reply 4: Thank you for your concerns. No, this is not part of 11087 images. We trained and validated the models using the 11807 images from **Shenzhen ophthalmic Centre** (please see line 254-258), and we further verified the DL models using 400 images from **Zhongshan ophthalmic Centre** (please see line 315-318). To make this point more clearly, **we clarified it in line 308**.

Comment 5: The authors categorize central serous chorioretinopathy as non-RD. But usually CSC manifested as shallow detachment of the macula. Did the model have difficult to identify

CSC from RD or macula-off RD?

Reply 5: Thank you for your question. Our models are capable of identifying CSC from RD and macular-off RD as we do not find this kind of misclassification in false-positive images. In addition, even if the models misclassify CSC as RD, this false-positive classification seems to be reasonable given that such eyes with CSC would benefit from further clinical investigation.

Comment 6: From figure 4, 5, 6, some images have shadow of eyelashes or eye lid. Did these shadows affect the model? How to minimize the interference ?

Reply 6: Thank you for your question. The shadows may affect the model, but this influence is small according to the high sensitivity of our DL models and the corresponding heatmaps. The best approach to minimize this interference is to help patients widen their eyes when taking UWF images.

Comment 7: The authors concluded that “this system can optimal timing for RD repair, and predict the potential vision prognosis”. This conclusion can’t be achieved from one-time fundus image without follow-up. The authors should amend their conclusion.

Reply 7: Thank you for your advice. We revised the conclusion as you suggested as follows: this system can automatically provide guidance to patients regarding appropriate preoperative posturing to reduce RD progression and the urgency of RD repair.

Changes in the text: see line 30 to 32; line 249-250.